# The Diagnostic Value of Anti-Parietal Cell and Intrinsic Factor Antibodies, Pepsinogens, and Gastrin-17 in Corpus-Restricted Atrophic Gastritis

**DOI:** 10.3390/diagnostics12112784

**Published:** 2022-11-14

**Authors:** Petra Kriķe, Zakera Shums, Inese Poļaka, Ilze Kikuste, Aigars Vanags, Ivars Tolmanis, Sergejs Isajevs, Inta Liepniece-Karele, Daiga Santare, Lilian Tzivian, Dace Rudzīte, Minkyo Song, M. Constanza Camargo, Gary L. Norman, Mārcis Leja

**Affiliations:** 1Institute of Clinical and Preventive Medicine & Faculty of Medicine, University of Latvia, LV-1586 Rīga, Latvia; 2Pauls Stradins Clinical University Hospital, LV-1002 Riga, Latvia; 3Headquarters & Technology Center Autoimmunity, Werfen, San Diego, CA 92121, USA; 4Digestive Diseases Centre GASTRO, Hong Kong 122001, China; 5Academic Histology Laboratory, LV-1073 Riga, Latvia; 6Department of Pathology, Rīga Stradiņš University, LV-1007 Riga, Latvia; 7Riga East University Hospital, LV-1038 Riga, Latvia; 8Division of Cancer Epidemiology and Genetics, National Cancer Institute, Rockville, MD 20850, USA

**Keywords:** autoimmune gastritis, atrophic gastritis, corpus-restricted atrophic gastritis, anti-parietal cell antibodies, intrinsic factor antibodies, pepsinogens, gastrin-17

## Abstract

We aimed to determine the diagnostic value of anti-parietal cell antibodies (anti-PCA), anti-intrinsic factor antibodies (anti-IFA), pepsinogen ratio (PGI/II), and gastrin-17 (G-17) in corpus-restricted atrophic gastritis (CRAG) detected by ELISA (Inova, Biohit). Our study compared 29 CRAG cases against 58 age- and sex-matched controls with mild or no atrophy. Anti-PCA and anti-IFA positive cutoff values were ≥25 units for both. PGI/II value <3 was considered characteristic for atrophy; positive cutoff values for G-17 and anti-*H. pylori* IgG were >5 pg/L and >30 EIU. Anti-PCA was positive in 65.5% For CRAG cases and 13.8% of the controls (*p* < 0.0001), anti-IFA was positive in 13.8% and 0% (*p* = 0.01), respectively. Decreased pepsinogen levels were present in 79.3% of CRAG cases and 10.3% of the controls (*p* < 0.0001). PGI/II ratio was the best single biomarker, with sensitivity = 79%, specificity = 90%, and AUC 0.90. The combined use of PGI/II and anti-PCA resulted in AUC 0.93 for detecting CRAG. Our study suggests that the best combination of non-invasive biomarkers for detecting CRAG is PGI/II with anti-PCA. The addition of G-17 and anti-IFA is of little utility in clinical application.

## 1. Introduction

Stomach infection with *Helicobacter pylori* and autoimmunity are the key mechanisms in the development of atrophic gastritis, a precancerous lesion for gastric adenocarcinoma [1,2]. As the prevalence of autoimmune gastritis appears to be increasing globally, proper diagnosis of this condition is becoming more important. For example, the prevalence of autoimmune corpus gastritis has increased over the past several decades, especially among younger adults in Northern Sweden [3]. The association of anti-parietal cell antibodies (anti-PCA) and anti-intrinsic factor antibodies (anti-IFA) with the risk of gastric cancer among Finnish women of reproductive age and older men was recently studied [4]. The study concluded that pre-diagnostic seropositivity to anti-PCA was associated with an elevated risk of gastric cancer among younger women, but not among older men.

Corpus-restricted atrophic gastritis (CRAG), i.e., the presence of evident atrophy in the corpus part of the stomach without atrophic lesions in the antrum, is predominantly the result of an autoimmune process where anti-PCA production destroys gastric parietal cells involved in intrinsic factor production. This further results in an inability to absorb vitamin B12, and leads to pernicious anemia and an increased risk for gastric cancer [5,6,7]. CRAG patients have three-fold increased risk of gastric cancer and a thirteen-fold increased risk of carcinoids compared to the general population [8]. Pernicious anemia carries a 2.84 relative risk increase for gastric cancer, according to a recent meta-analysis for autoimmune diseases associated with gastric cancer risk [9]. 

Early detection of CRAG could contribute to a timelier diagnosis of pernicious anemia and gastric cancer. Currently, the diagnosis of CRAG requires a combination of clinical, serological, and histopathologic data, where biopsy is the most reliable method to evaluate the presence of atrophic gastritis [10]. 

Increasing the understanding of a particular biomarker’s correlation with pathology-based CRAG diagnosis could further improve the identification of patients at risk and guide diagnostic and treatment activities [1]. Anti-PCA is a diagnostic marker of autoimmune gastritis, and is also a predictive marker of subsequent corpus atrophy and pernicious anemia [11]. Anti-IFA is a biomarker of pernicious anemia and appears at later stages of atrophic gastritis [12]. Serum pepsinogen ratio (PGI/II) ≤ 3 has been linked to the diagnosis of atrophic gastritis [13]. Gastrin-17 (G-17) is expected to be high in CRAG as a result of the compensatory mechanism when hydrochloric acid secretion in the corpus decreases due to parietal cell depletion but atrophy in the antrum is lacking, a situation in which production of G-17 increases [14]. PGI/II and G-17 are well-established biomarkers for gastric atrophy with a high correlation with gastric biopsy [15,16]. All of the above-listed markers have been reported to be of value for the detection of atrophic autoimmune gastritis; however, we are lacking clear diagnostic algorithms based on the routinely available serological tests. A recent guideline for the management of epithelial precancerous conditions and lesions in the stomach (MAPS II) acknowledges that most studies describing the link between autoimmune gastritis and gastric cancer had included patients based on low B12 levels rather than histologically confirmed diagnosis [17,18]. MAPS II guidelines recommend that patients with autoimmune gastritis may benefit from endoscopic follow-up every 3–5 years. Regarding noninvasive assessment, it is advised to use PGI serum levels and/or a low PGI/II to identify patients with advanced stages of atrophic gastritis as candidates for endoscopy. Further studies exploring other potential biomarkers or combinations to identify atrophic gastritis patients are required. There is no classification of gastritis that is universally recognized. In order to assess the severity of gastric atrophy and intestinal metaplasia, OLGA and OLGIM staging systems have been used in recent years [19,20].

Heterogeneity of the study groups, resulting from the inclusion of patients with atrophy either due to autoimmunity or *H. pylori* infection and/or the lack of endoscopic data, has been the major limiting factor for the biomarker evaluation of previous studies. To overcome this, we have evaluated the value of routinely available markers, namely anti-PCA, anti-IFA, PGI/II, and G-17, in a well-characterized group of patients with CRAG, therefore pre-selecting a group of patients with a very high likelihood of atrophic gastritis development as a result of autoimmune mechanisms.

## 2. Materials and Methods

### 2.1. Study Population

Study individuals with CRAG were selected from a larger group of 1978 patients being referred for upper endoscopy due to dyspeptic symptoms at the Digestive Diseases Centre GASTRO. Each of the patients diagnosed with CRAG was age and sex-matched to two controls, both with mild symptoms or without any signs of histological atrophy. Patients with gastric cancer were not included in this study. 

### 2.2. Histology

CRAG was defined as the presence of moderate-to-severe atrophy and any stage of intestinal metaplasia in the corpus, with the absence of moderate-to-severe atrophy or any stage of intestinal metaplasia either in the antral part or incisura. Controls could have no atrophy or mild atrophy, but no intestinal metaplasia at any location. To test for this, five biopsies were collected and analyzed according to the updated Sydney system [21]. Haematoxylin and eosin staining was used as the routine procedure. Two pathologists reviewed each biopsy, and if there was a discrepancy between their assessments, a consensus review was performed. The presence of *H. pylori* was evaluated based on modified Giemsa staining. *H. pylori* was considered positive if the infection was reported in any of the biopsies.

### 2.3. Serology

Blood samples were obtained after an overnight fast and processed immediately thereafter; frozen plasma samples were stored at −70 °C until analyzed. Enzyme-linked immunosorbent assay (ELISA) was used for biomarker analysis. Anti-PCA and anti-IFA were analyzed by QUANTA Lite^®^ ELISA kits (Inova Diagnostics Inc., San Diego, CA, USA)—the absorbance was measured using a BioTek [ELx808] reader in the laboratory of Inova Diagnostics Inc., San Diego, CA, USA. Samples sent to the USA were shipped on dry ice; their identities were blinded, and completed test results were sent to PK and ML for unblinding and analysis. PGI and II, G-17, and anti-*H. pylori* IgG ELISA (GastroPanel, Biohit, Oyj., Finland) were tested using an automated two-microwell plate analyser (Personal-Lab, Adaltys, Italy), and regular QC was performed with high- and low-range control sample materials provided by the manufacturer. The cutoff values recommended by the manufacturers were used. Anti-PCA and anti-IFA antibody values of ≥25 units were considered positive, values between 20.1 and 24.9 units were considered equivocal, and values ≤ 20 units were considered negative [22]. PGI/II ratio values below 3 were considered decreased and characteristic of corpus atrophy [4]. G-17 values > 5.0 pmol/L for fasting specimens were considered to be increased and attributed to corpus atrophy, and anti-*H. pylori* IgG values of ≥30 units were considered positive.

### 2.4. Statistical Analysis 

We obtained descriptive statistics of the study population and biomarker values. The differences between the mean biomarker values in the CRAG and control groups were calculated using the Wilcoxon signed-rank test. We calculated the sensitivity, specificity, positive predictive value (PPV), negative predictive value (NPV), and area under the receiver operating characteristic curve (AUC). Conditional logistic regression models were used to calculate odds ratios (OR) for each biomarker and the corresponding 95% confidence interval. Venn diagrams were created by Venny version 2.1 [23]. The Pearson correlation was run to assess the relationship among all measured biomarkers. Statistical analysis was performed by SPSS version 22.0 and SAS (version 9.4).

## 3. Results

Altogether, 29 individuals with CRAG were identified and matched to 58 controls with mild or no atrophy. The description of the sociodemographic variables of the study groups is shown in Table 1. The groups were well-balanced for age, height, and weight. In total, 28 CRAG cases and 55 controls had not received proton pump inhibitors (PPIs) for the previous month prior to enrollment, and 25 CRAG cases and 36 controls had not used PPIs during the last year. Data on PPI use were missing for one CRAG case. Eight CRAG cases and eighteen controls reported previous *H. pylori* eradication treatment, and for one CRAG case, eradication status was not clear. *H. pylori* seropositivity was not different between cases and controls (51.7% vs. 67.2%; *p* = 0.16). The mean levels of anti-PCA, anti-IFA, G-17, and PGI/II ratio results and the seroprevalence of these biomarkers are available in Table 2. Sensitivity, specificity, PPV, and NPV for CRAG patients for individual biomarkers and the best combination are shown in Table 3. AUC results were calculated for anti-PCA, anti-IFA, gastrin-17, and PGI/II ratio (Table 4). Combined biomarker G-17, PGI/II ratio, anti-PCA, and anti-IFA area under the ROC ^a^ curve results are also provided (Table 5).

The overlap of positive biomarker values G-17, PGI/II ratio, anti-PCA, and anti-IFA in CRAG patients and controls is illustrated by the Venn diagrams in Figure 1. In total, 96.7% (28 out of 29) of patients with CRAG had at least one positive biomarker value, and in four patients (13.8%) all four biomarkers were positive. The maximum positive value overlap was observed for G-17, PGI/II ratio, and anti-PCA in 10 (34.5%) CRAG patients (*n* = 29). In the control group, 20 (34.5%) of the 58 patients had at least one positive biomarker value; no patients had all four biomarkers positive. Simultaneous positivity for G-17, PGI/II ratio, and anti-PCA antibody was observed only in four (2.9%) control group patients (*n* = 58).

Pearson correlation coefficient values for anti-PCA, anti-IFA, gastrin-17, and PGI/II ratio for all patients and CRAG and control groups separately were calculated (Table 6). There was a moderate negative correlation between G-17 and PGI/II ratio (rho = −0.61, *p* < 0.0001). There was a low positive correlation between G-17 and anti-PCA (rs = 0.48, *p* < 0.0001) and between anti-IFA and G-17 (rho = 0.45, *p* < 0.0001); a medium negative correlation between PGI/II ratio and anti-PCA (rho = −0.38, *p* = 0.0003); and a low correlation between anti-IFA and PGI/II ratio (rho = −0.21, *p* =0.0474). No statistically significant correlation was found between anti-IFA and anti-PCA. The correlation was significant at the 0.01 level (2-tailed).

## 4. Discussion

Our study focused on gastritis patients whose diagnosis was based on histology assessment. To avoid as much as possible the potential effects related to *H. pylori*-caused atrophy as opposed to autoimmunity-caused atrophy, we selected the CRAG group, since the role of autoimmunity would be expected to be prevailing in this subgroup; we excluded patients with both antrum and corpus atrophy, where the significant impact could be made by *H. pylori* infection. To our knowledge, previous studies were lacking such a strict preselection. For the controls, study individuals who were not expected to have gastric autoimmunity according to the pathology results were identified. Therefore, our study was able to judge the performance of the biomarkers as precisely as possible for the detection of autoimmune gastritis with atrophy—the risk condition for developing gastric cancer.

Anti-PCA and anti-IFA are well-established biomarkers used for laboratory diagnosis of autoimmune gastritis [11]. Many studies on anti-PCA levels in patients with CRAG have used pernicious anemia patients as a proxy for severe total gastric mucosa atrophy. Anti-PCA is positive in 70–80% of pernicious anemia patients, but it also is positive in 7–10% of the healthy population, so the test is not highly specific for pernicious anemia patients [24,25,26]. A positive anti-IFA test has a high positive predictive value (95%) for the presence of pernicious anemia [27] with a concurrent low false positive rate (1–2%), indicating a high specificity. However, positivity for anti-IFA has been observed in only 40–60% of pernicious anemia patients, so the lack of anti-IFA does not rule out the diagnosis [26,28,29]. 

There are different practices for the clinical use of these biomarkers in different countries. Limited guidelines on atrophic gastritis are available. The recent American Gastroenterological Association Clinical Practice Update on the Diagnosis and Management of Atrophic Gastritis advises that in patients with histology compatible with autoimmune gastritis, both anti-PCA and anti-IFA should be checked to assist with the diagnosis [30]. Meanwhile, guidelines for the diagnosis and treatment of cobalamin and folate disorders of the British Society for Hematology suggest that all patients suspected of having pernicious anemia should be tested for anti-IFA, while anti-PCA antibody testing for diagnosing pernicious anemia is not recommended [26].

In our study, we also found high sensitivity in anti-PCA and low sensitivity in anti-IFA for detecting CRAG, similarly to previous studies. Anti-PCA, as a single marker with an AUC value of 0.76, had a lower discriminatory ability as compared to PGI/II, and was close to G-17 for diagnosing CRAG. Anti-IFA had a very low sensitivity of 14%, but a very high specificity of 100%, in line with prior studies [26]. A recent study reported specificity and PPV of 100% for patients with overt or latent pernicious anemia using the same assay as employed in our study [22]. Positive anti-IFA may appear at later stages of gastric atrophy for patients who have initially been anti-PCA positive, as described by Ottesen et al. [31].

In a study of subjects who were young nonanemic blood donors, predominantly male, and at low risk of atrophic gastritis, the prevalence of anti-PCA was 7.8% detected by ELISA [32]. Considering that some patients with signs of gastric autoimmunity characterized by anti-PCA may not develop atrophy, and therefore may not be at increased risk for gastric cancer, we have evaluated the rationale of combining anti-PCA and anti-IFA (the latter highly specific for atrophic gastritis) as markers of autoimmunity with the serological markers for gastric mucosal atrophy, namely pepsinogen and G-17 detection.

As expected, our study has demonstrated decreased PGI/II in the majority of CRAG cases. G-17 was increased in most of the patients with CRAG, but not in all. A negative correlation was found between PGI/II ratio and G-17. Corpus atrophy may have decreased pepsinogen I synthesis, thereby reducing hydrochloric acid secretion and inducing compensatory increased G-17 production in the antrum. These findings had been previously observed in older studies [13]. Normal G-17 levels in a subset of patients with CRAG may be explained by continued gastric acid output even in the presence of corpus atrophy, where acid secretion has negative feedback on gastrin levels. Increased gastrin levels in the control group may be a result of several factors, including *H. pylori* infection and PPI administration [33]. We would have expected that all the GRAG group patients should have increased G-17 values, as they had significant atrophy in the corpus, but lack of atrophy in antrum in the antrum. We found a high sensitivity with lower specificity with G-17 in detecting CRAG; therefore, we find the use of this biomarker of very limited use. 

PGI/II ratio had a high PPV and a high NPV for CRAG. The PGI/II ratio also had the highest discriminatory value among the biomarkers, demonstrating the strongest potential for a single marker to identify patients with CRAG.

Our group previously studied the validity of PGI/II and G-17 in detecting gastric atrophy, and our new results are comparable to the previously published ones [34,35,36]. 

Combining anti-PCA and PGI/II demonstrated increased discriminatory ability with AUC 0.93 as compared to single biomarkers. Adding G-17 and/or anti-IFA to the combination did not result in a higher AUC.

Our group of patients had relatively high prevalence of *H. pylori* infection, i.e., 63.0% in the CRAG group and 45.5% in the control group, according to the histology. Although in a typical case of autoimmune gastritis, *H. pylori* infection could be expected to be absent, association of autoimmunity and *H. pylori* infection is also characteristic [37]. This overlap is of high likelihood, since about 50% of the global population is affected by *H. pylori* [38], and a high prevalence of the infection has also been reported in Latvia [39]. Several limitations of our study are acknowledged. We acknowledge that the total number of patients was relatively low; at the same time, according to our knowledge, this is the only study, and therefore the largest study so far published with such exact patient recruitment criteria. Since the use of PPIs is known to interfere with the results of G-17, we intended to exclude all patients who had recently used PPIs from the study; however, some of these patients were included in each of the groups. Eight CRAG cases that reported previous *H. pylori* eradication treatment and one CRAG case where eradication status was not clear were calculated as one subgroup. Five patients had histology information missing on *H. pylori* positivity. However, most importantly, we cannot exclude that in some of the patients with CRAG, the lesions could have developed as the result of *H. pylori* infection or a combination of *H. pylori* infection and autoimmunity. We believe, however, that this is extremely unlikely, since in *H. pylori*-related atrophy, the predominant lesions would have been found in the antral part of the stomach. To avoid the influence of *H. pylori* infection, patients with panatrophy (atrophy in either corpus or antrum) were not included in the analysis.

## 5. Conclusions

Our study suggests that the best combination of non-invasive biomarkers for detecting CRAG is the combination of PGI/II and anti-PCA biomarkers. The addition of G-17 and anti-IFA appeared to add little useful value for clinical detection of CRAG, although there could be a role for these biomarkers in biomarker panels in the future.

## Figures and Tables

**Figure 1 diagnostics-12-02784-f001:**
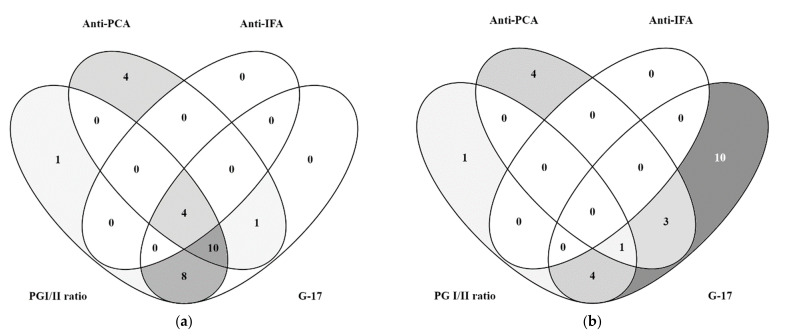
Venn diagrams illustrating patient numbers with positive overlapping biomarker values G-17, PGI/II ratio, anti-PCA, and anti-IFA. (**a**) From 29 CRAG cases, 28 had at least one positive biomarker value. One patient had all values negative. (**b**) From 58 controls, 20 controls had at least one positive biomarker value.

**Table 1 diagnostics-12-02784-t001:** Description of sociodemographic variables and *H. pylori* status in corpus-restricted atrophic gastritis (CRAG) patients and controls.

	CRAG Cases (*n* = 29)	Controls (*n* = 58)	*p*-Value
Age, mean, ±SD ^a^	57 ± 13	57 ± 13	1.00
Female, *n* (%)	21 (72.4)	42 (72.4)	1.00
Height (cm), mean, ±SD	168.8 ± 7.4	167.7 ± 8.6	0.57
Weight (kg), mean, ±SD	74.3 ± 15.7	73.8 ± 17.9	0.89
*H. pylori* positivity (based on serology) *n* (%)	15 (51.7)	39 (67.2)	0.16
*H. pylori* positivity (based on histology) *n* (%)	17 (63.0)	25 (45.5)	0.14
*H. pylori* eradication (self-reported) *n* (%)	9 ^b^ (31.0)	18 (31.0)	1.00

^a^ SD—standard deviation. ^b^ For 1 patient, eradication status was not clear.

**Table 2 diagnostics-12-02784-t002:** Levels of gastrin-17, anti-parietal cell antibodies (anti-PCA), anti-intrinsic factor antibodies (anti-IFA), and pepsinogen ratio (PGI/II) in corpus-restricted atrophic gastritis (CRAG) patients and controls; seroprevalence of biomarkers in CRAG and controls.

Biomarker Values	CRAG Cases (*n* = 29)	Controls (*n* = 58)	*p*-Value ^d^
Levels, mean ±SD ^a^			
G-17 pg/L	40.4 ± 29.7	6.2 ± 11.7	<0.0001
PGI/II	2.4 ± 2.7	8.8 ± 4.2	<0.0001
Anti-PCA Units	52.7 ± 46.0	9.5 ± 16.5	<0.0001
Anti-IFA Units	14.7 ± 25.2	5.2 ± 3.3	0.05
Seropositivity, *n* (%)			
High G-17 ^b^	23 (79.3)	18 (31.0)	<0.0001
Low PGI/II ^c^	23 (79.3)	6 (10.3)	<0.0001
Anti-PCA	19 (65.5)	8 (13.8)	<0.0001
Anti-IFA	4 (13.8)	0 (0.0)	0.01

^a^ SD—standard deviation. ^b^ High G-17 cut-off value > 5. ^c^ Low PGI/II ratio cut-off value. ^d^ Differences between CRAG and control based on *t*-test for continuous values and chi-square/Fisher’s exact test for categorical values.

**Table 3 diagnostics-12-02784-t003:** Biomarker gastrin-17, anti-PCA, anti-IFA and PGI/II ratio sensitivity, specificity, positive predictive value, and negative predictive value for CRAG diagnosis.

	Sensitivity			Specificity			PPV			NPV		
Biomarker	Estimate	Lower CI	Upper CI	Estimate	Lower CI	Upper CI	Estimate	Lower CI	Upper CI	Estimate	Lower CI	Upper CI
G-17	0.79	0.65	0.94	0.69	0.57	0.81	0.56	0.41	0.71	0.87	0.77	0.97
PGI/II	0.79	0.65	0.94	0.90	0.82	0.97	0.79	0.65	0.94	0.90	0.82	0.97
Anti-PCA	0.66	0.48	0.83	0.85	0.75	0.94	0.68	0.51	0.85	0.83	0.73	0.93
Anti-IFA ^a^	0.14	0.04	0.32	1.00	0.94	1.00	1.00	0.40	1.00	0.70	0.59	0.79
*H. pylori* serology	0.52	0.34	0.70	0.33	0.21	0.45	0.28	0.16	0.40	0.58	0.41	0.74
*H. pylori* histology	0.63	0.45	0.81	0.55	0.41	0.68	0.40	0.26	0.55	0.75	0.62	0.88
PGI/II + Anti-PCA	0.97	0.90	1.00	0.78	0.67	0.88	0.68	0.54	0.83	0.98	0.94	1.00

^a^ Exact method was used to assess confidence intervals (CI).

**Table 4 diagnostics-12-02784-t004:** Biomarker gastrin-17, anti-PCA, anti-IFA, and PGI/II ratio area under the ROC ^a^ curve results based on levels and category of seropositivity.

Biomarker Values	Area under the ROC Curve	SE ^b^	*p*-Value ^c^	Asymptotic 95% Confidence Interval	
				Lower Bound	Upper Bound
**Levels**					
G-17	0.85	0.05	<0.0001	0.75	0.95
PGI/II ratio	0.90	0.03	<0.0001	0.84	0.97
Anti-PCA	0.79	0.06	<0.0001	0.68	0.91
Anti-IFA	0.58	0.07	0.23	0.45	0.72
**Seropositivity**					
G-17	0.74	0.05	<0.0001	0.65	0.84
PGI/II ratio	0.85	0.04	<0.0001	0.76	0.93
Anti-PCA	0.76	0.05	<0.0001	0.66	0.86
Anti-IFA	0.57	0.03	0.03	0.51	0.63

^a^ ROC, Receiver Operating Characteristic; SE, standard error. ^b^ Under the nonparametric assumption. ^c^ Null hypothesis: true area = 0.5.

**Table 5 diagnostics-12-02784-t005:** Combined biomarker G-17, PGI/II ratio, anti-PCA and anti-IFA area under the ROC ^a^ curve results.

Combined Biomarkers				AUC ^b^	SE ^c^	*p*-Value	LCI ^d^	UCI ^e^
G-17				0.74	0.05	<0.001	0.65	0.84
PGI/II				0.85	0.04	<0.001	0.76	0.93
Anti-PCA				0.76	0.05	<0.001	0.66	0.86
Anti-IFA				0.57	0.03	0.03	0.51	0.63
G-17	PGI/II			0.84	0.05	<0.001	0.75	0.94
G-17	Anti-PCA			0.85	0.04	<0.001	0.77	0.93
G-17	Anti-IFA			0.76	0.05	<0.001	0.67	0.86
PGI/II	Anti-PCA			0.93	0.03	<0.001	0.87	0.98
PGI/II	Anti-IFA			0.85	0.04	<0.001	0.77	0.94
G-17	PGI/II	Anti-PCA		0.93	0.03	<0.001	0.88	0.98
G-17	PGI/II	Anti-IFA		0.85	0.05	<0.001	0.75	0.94
G-17	Anti-PCA	Anti-IFA		0.85	0.04	<0.001	0.77	0.93
G-17	PGI/II	Anti-PCA	Anti-IFA	0.93	0.03	<0.001	0.88	0.98

^a^ ROC, Receiver Operating Characteristic. ^b^ AUC. ^c^ SE Standard error. ^d^ LCI Confidence interval lower bound. ^e^ UCI Confidence interval upper bound.

**Table 6 diagnostics-12-02784-t006:** Pearson correlation coefficients illustrating correlations between biomarker values G-17, PGI/II ratio, anti-PCA, and anti-IFA in all individuals, CRAG cases, and controls ^a^.

All Patients (*n* = 87)						
	G-17	PGI	PGII	PGI/II	Anti-PCA	Anti-IFA
PGI	−0.44 ^b^					
PGII	0.01	0.58 ^b^				
PGI/II	−0.61 ^b^	0.53 ^b^	−0.24 ^c^			
Anti-PCA	0.48 ^b^	−0.31 ^c^	−0.06	−0.38 ^c^		
Anti-IFA	0.45 ^b^	−0.21 ^c^	−0.08	−0.21 ^c^	0.13	
HP IgG	−0.17	0.03	0.28 ^c^	−0.15	−0.19	−0.12
CRAG cases (*n* = 29)						
	G-17	PGI	PGII	PGI/II	Anti-PCA	Anti-IFA
PGI	−0.50 ^c^					
PGII	−0.05	0.54 ^c^				
PGI/II	−0.61 ^c^	0.67 ^b^	−0.06			
Anti-PCA	0.24	−0.15	−0.20	−0.01		
Anti-IFA	0.45 ^c^	−0.13	−0.09	−0.14	−0.06	
HP IgG	−0.37 ^c^	0.27	0.33	0.03	−0.37 ^c^	−0.19
Control cases (*n* = 58)						
	G-17	PGI	PGII	PGI/II	Anti-PCA	Anti-IFA
PGI	−0.15					
PGII	0.01	0.72 ^b^				
PGI/II	−0.31 ^c^	0.31 ^c^	−0.35 ^c^			
Anti-PCA	−0.01	−0.08	0.00	−0.07		
Anti-IFA	−0.20	−0.30 ^c^	−0.34 ^c^	0.08	0.09	
HP IgG	−0.08	−0.03	0.25	−0.27 ^c^	−0.11	−0.12

^a^ Prob > |r| under H0: Rho = 0. ^b^ *p* < 0.0001. ^c^ *p* < 0.05.

## Data Availability

There is no supplementary online information or other data published in public domains available for this study.

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
