# Peer review of "The Diagnostic Value of Anti-Parietal Cell and Intrinsic Factor Antibodies, Pepsinogens, and Gastrin-17 in Corpus-Restricted Atrophic Gastritis"

_diagnostics, 2022, doi:10.3390/diagnostics12112784_

Round 1
Reviewer 1 Report
The paper "The Diagnostic Value of Anti-Parietal Cell and Intrinsic Factor Antibodies, Pepsinogens, and Gastrin-17 in Corpus Restricted Atrophic Gastritis" is extremely up-to-date, as there are many discussion about the diagnosis of CRAG, gastritis that can induce the gastric cancer. The methodology is sound, the results are presented in very well interpreted tables, the discussions are covering the topic. The conclusions are coherent to the data obtained in the study.
However, I would suggest more discussion about the HP, as approximately half of the patients had also the infection.
Reviewer 2 Report
The current research shows important relationship of serum anti-PCA, anti-IFA, PG, G-17 and Hp-IgG with gastric atrophy, which is cancer related. The style and design are OK and the results reliable. There are some minor issues to be concerned.
1.H.pylori's positive rate by serum and pathology is opposite in CRAG and control group. Do we need UBT for the tests?
2.The sample size is not big enough.
3. Some reports on PG, G-17 and Hp were missed in the literature, including but not limited to Gut 2019;68:1576–1587, Gut 2021;70:829–837., Gastroenterology Research and Practice Volume 2022, Article ID 7639968, Gastric Cancer (2021) 24:1194–1202, etc.
